# Engineering Bio-Adhesives Based on Protein–Polysaccharide Phase Separation

**DOI:** 10.3390/ijms23179987

**Published:** 2022-09-01

**Authors:** Zoobia Bashir, Wenting Yu, Zhengyu Xu, Yiran Li, Jiancheng Lai, Ying Li, Yi Cao, Bin Xue

**Affiliations:** 1Collaborative Innovation Center of Advanced Microstructures, National Laboratory of Solid State Microstructure, Key Laboratory of Intelligent Optical Sensing and Manipulation, Ministry of Education, Department of Physics, Nanjing University, Nanjing 210093, China; 2Oujiang Laboratory (Zhejiang Lab for Regenerative Medicine, Vision and Brain Health), Wenzhou Institute, University of Chinese Academy of Sciences, Wenzhou 325001, China; 3Institute of Advanced Materials and Flexible Electronics (IAMFE), School of Chemistry and Materials Science, Nanjing University of Information Science & Technology, Nanjing 210044, China

**Keywords:** bio-adhesives, phase separation, tissue adhesion, wet adhesion, dopa

## Abstract

Glue-type bio-adhesives are in high demand for many applications, including hemostasis, wound closure, and integration of bioelectronic devices, due to their injectable ability and in situ adhesion. However, most glue-type bio-adhesives cannot be used for short-term tissue adhesion due to their weak instant cohesion. Here, we show a novel glue-type bio-adhesive based on the phase separation of proteins and polysaccharides by functionalizing polysaccharides with dopa. The bio-adhesive exhibits increased adhesion performance and enhanced phase separation behaviors. Because of the cohesion from phase separation and adhesion from dopa, the bio-adhesive shows excellent instant and long-term adhesion performance for both organic and inorganic substrates. The long-term adhesion strength of the bio-glue on wet tissues reached 1.48 MPa (shear strength), while the interfacial toughness reached ~880 J m^−2^. Due to the unique phase separation behaviors, the bio-glue can even work normally in aqueous environments. At last, the feasibility of this glue-type bio-adhesive in the adhesion of various visceral tissues in vitro was demonstrated to have excellent biocompatibility. Given the convenience of application, biocompatibility, and robust bio-adhesion, we anticipate the bio-glue may find broad biomedical and clinical applications.

## 1. Introduction

Bio-adhesives is of great importance for surgical suturing and internal monitoring due to the advantages of simpler deployment, time-saving operation, and prevention of related complications compared to traditional stitches [1,2,3]. Ideal bio-adhesives need to achieve rapid, robust, conformal integration with various wet dynamic tissues or inorganic devices [4], which require strong and fast interfacial adhesion. Moreover, the intrinsic strength of bio-adhesives also needs to be strong enough to resist the stress generated by the adhered objects and hold the conjunction of different tissues or substrates, which requires strong internal cohesion. Generally, bio-adhesives can be divided into glue-type and tape-type strategies. Compared to tape-type bio-adhesives [5,6,7,8], glue-type adhesives are especially suitable for in situ adhesion due to their injectable ability and localized curing progress [9,10,11]. However, glue-type adhesives usually cannot be applied for instant tissue adhesion due to their weak cohesion strength [12,13]. Although numerous efforts have been made to explore bio-adhesives [14,15,16], such as cyanoacrylate derivatives [11], fibrin glues [17], and 3, 4-dihydroxyphenylalanine (Dopa)-based adhesives [18,19,20,21,22], building glue-type bio-adhesives suitable for tissue adhesions still remains challenging.

Recently, it has been reported that phase separation plays an important role in natural bio-adhesive systems [23,24,25]. For example, foot protein 5 (Pvfp-5) from mussels can undergo liquid–liquid phase separation during underwater adhesion [26,27]. The phase separation of protein can significantly improve the strength of cohesion and speed up the process of internal curing in bio-adhesives. Meanwhile, a post-translated amino acid l-3,4-dihydroxyphenylalanine (Dopa)-containing dihydroxyl phenylalanine was widely reported to contribute to the water-resistant adhesive ability of mussels [28,29,30]. Dopa can form dynamic interactions (charge–charge interactions, cation–π interactions, coordination, hydrogen bonding, and hydrophobic effects) [20,21,31,32] and permanent conjunctions (covalent bond between oxidized dopa and amino/thiol groups) [33] with various substrates. Interestingly, it has been found in biological systems that proteins and polysaccharides can form electrostatic complexes and coacervates, thus exhibiting phase separation behaviors [34,35,36]. These complexes, or coacervates, have many applications in the food and biomedical fields, including fat substitution [37], protein purification [38], and drug delivery [39,40]. However, the exploration of bio-adhesives based on protein–polysaccharide phase separation has rarely been reported. Taking advantage of cohesion from protein–polysaccharide phase separation and the advantage of adhesion from dopa, we should be able to engineer glue-type bio-adhesives with fast and strong adhesive performance as well as high biocompatibility.

In this work, we used pectin-β-lactoglobulin phase separation coupled with dopa as a model system to demonstrate this concept of building glue-type bio-adhesives. By modifying pectin with dopa, we developed a bio-adhesive based on the phase separation of pectin and β-lactoglobulin (the glue-type bio-adhesive is denoted as bio-glue hereafter). The resulting bio-glue exhibited enhanced phase separation behaviors as well as increased adhesion performance. The long-term adhesion strength of the bio-glue on wet tissues reached 1.48 MPa (shear strength), while the interfacial toughness reached ~ 880 J m^−2^, outperforming many reported bio-adhesives [41,42,43,44,45,46]. More importantly, the bio-glue showed an increased instant adhesion strength of 0.27 MPa in 1 min, making it suitable for fast and strong adherence of irregular tissues. We further demonstrate the feasibility of this bio-glue in the adhesion of various visceral tissues in vitro with unaffected biocompatibility. We anticipate that the design principle represents a new general approach to build glue-type tissue adhesives and that such bio-adhesives can find broad application in the fields of biomedical and clinical medicine.

## 2. Results and Discussion

### 2.1. Design of the Glue-Type Bio-Adhesive Based on Protein–Polysaccharide Phase Separation

As illustrated in Figure 1, the glue-type bio-adhesive is formed by dopa-modified pectin (pectin-dopa) and β-lactoglobulin (Figure 1). Driven by the long-range character of the electrostatic interaction between β-lactoglobulin and oppositely charged pectin, soluble pectin-dopa and β-lactoglobulin aggregate into insoluble protein–polysaccharide complexes, leading to phase separation in solutions. The phase separation performance ensured that bio-adhesives would not dissociate or be diluted even in wet conditions. In addition, the coacervation of pectin and β-lactoglobulin can enhance cohesive strength during instant adhesion. Dopa at the side chain acted as the functional group to improve the surface adhesion between the bio-adhesive and various substrates. During typical adhesion, dopa forms dynamic and fast junctions with tissues or substrates via ionic interactions, cation–π interactions, and hydrogen bonds, ensuring the stability of short-term adhesion [31,47]. Then, dopa is oxidized into dopaquinone and gradually forms covalent bonds with amino/thiol groups through Michael addition. Moreover, the covalent bond between dopaquinone and amino acids from pectin or β-lactoglobulin, as well as the aggregation of hydrophobic dopaquinone, provide additional cohesion. All these mechanisms lead to the further enhancement of the adhesion strength. The bio-glue containing dopa-functionalized pectin β-lactoglobulin is hereafter named PD-L bio-glue. For comparison, bio-glue containing pectin β-lactoglobulin and pectin or pectin-dopa alone (named P-L bio-glue or PD bio-glue, respectively) were used as the control groups.

### 2.2. Phase Separation Based on Pectin-Dopa and β-Lactoglobulin

First, we investigated the phase separation of the bio-glue based on pectin-dopa and β-lactoglobulin. The synthesis of pectin-dopa was achieved through the reaction of dopamine and pectin under the catalysis of NHS and EDC. The reaction efficiency of pectin and dopa reached 11.8, 16.1, and 28.5% by adjusting the mass ratio of dopa and pectin (1:10, 1:2, and 1:1) during the synthesis (Appendix A). The chemical shift form H nuclear magnetic resonance (^1^H-NMR) spectrum at 6.5–6.9 which belongs to H of benzene further indicated that dopa was successfully connected with pectin (Appendix A). As shown in Figure 2A, obvious coacervates were observed in PD-L and P-L bio-glue, while no coacervates were observed in PD bio-glue, indicating that the introduction of dopa into pectin would not affect the phase separation of pectin-dopa and β-lactoglobulin. Moreover, the area of the coacervates in PD-L bio-glue was larger than that in P-L, and the distance between the adjacent coacervates in PD-L bio-glue was smaller (Figure 2B), suggesting that the modification of pectin with dopa can further improve the phase separation of pectin-dopa and β-lactoglobulin. The coacervate diameters of bio-glues were also quantitatively evaluated using dynamic light scattering (DLS). As indicated by Figure 2C, the coacervate diameter of PD-L was located in the range of 1.2~5.5 μm and concentrated at 4.6 μm. In contrast, the coacervate diameter of P-L was located in the range of 2.2~9.6 μm and concentrated at 3.9 μm, consistent with the results in the optical images. Furthermore, the rheological properties of the bio-glues were investigated. As shown in Appendix A, the storage modulus (G′) and loss modulus (G″) of the P-L bio-glue were 2410 and 1726 Pa and increased with the increasing frequency, indicating a typical response of polymers. In contrast, G′ and G″ of P-L bio-glue were 694 and 237 Pa and slightly increased in the frequency range of 1–10 Hz, suggesting enhanced dynamic stabilities. The high G′ and stable response to frequency of PD-L bio-glue indicated the enhanced cohesion inside the PD-L bio-glue, which is consistent with the observation of phase separation. G′ and G′ of pectin-dopa (G′ ~75 Pa; G″ ~37 Pa) was much smaller than those of PD-L and P-L, indicating the weaker cohesion. Moreover, the PD-L bio-glue showed a shear thinning property with the increase in frequency (Appendix A), suggesting that the bio-glue can be injected to the tissue defects directly in the clinic. The Fourier transform infrared (FT-IR) spectroscopy of PD-L, P-L, and PD were shown in Appendix A. The peaks at 3200–3500 cm^−1^ enhanced in the PD-L bio-glue comparing to that of P-L bio-glue, indicating the hydrogen bond and dopa oxidations.

Furthermore, the phase separation of PD-L bio-glue can be adjusted by varying the synthesis ratios of dopa and pectin (Appendix A) or the ratios of pectin-dopa and β-lactoglobulin (Appendix A). As shown in Appendix A, the diameter and average area of coacervates in PD-L bio-glue increased with the synthesis ratios of dopa and pectin, indicating that the phase separation of PD-L bio-glue at high synthesis ratios of dopa and pectin was improved. Moreover, Appendix A shows that the coacervate diameter reached a maximum at a pectin-dopa:β-lactoglobulin ratio of 1:1, and the spreading area of the coacervates was also higher than those at other ratios. All these results indicated that the highest efficiency of phase separation of pectin-dopa and β-lactoglobulin was achieved at a pectin-dopa:β-lactoglobulin ratio of 5:5 and synthesis ratios of dopa and pectin at 1:1.

### 2.3. Adhesion Performance

Next, the adhesion performance of the bio-glue was evaluated at different timescales. We used wet porcine skin as the model tissue because its mechanical and biological properties are similar to those of human skin [48]. We used two types of mechanical tests (lap shear test for shear strength and tensile test for tensile strength) to evaluate the adhesion performance of the bio-glue. As shown in Figure 3A–C and Appendix A, the shear strength of long-term adhesion (12 h) of porcine skin using PD-L bio-glue is higher than 0.5 MPa (1.48 MPa for skin–skin adhesion), which are larger than those of adhesions using P-L and PD bio-glue. Note that the shear strength for short-term adhesion (1 min) using PD-L bio-glue reached 0.27 MPa, which is outstanding compared to many commercial bio-adhesives [1,8]. A similar trend was observed for the tensile strength evaluation of adhesions. The tensile strength (Figure 3D–F and Appendix A) for short-term adhesion using PD-L bio-glue reached 0.22 MPa, and the tensile strength for long-term adhesion of PD-L bio-glue was more than 0.4 MPa (0.72 MPa for skins). As indicated by Appendix A, the interfacial toughness for the long-term adhesion of porcine skin using the PD-L bio-glue was ~880 J m^−2^, making it one of the toughest bio-adhesives [49,50,51,52,53]. In addition, the adhesion performance of the PD-L bio-glue on inorganic substrates and organic substrates was also studied. The shear and tensile strength for the long-term adhesion between glass substrates using PD-L bio-glue reached 0.40 and 0.54 MPa, respectively. For the long-term adhesion between porcine skin and glass substrates, the strength and tensile strength for the adhesion were 0.54 and 0.58 MPa, respectively. The adhesion strength after curing for different times (1 min, 1, 3, 6, 12, 18, and 24 h) was also studied to investigate the effects of dopamine oxidation kinetic on the adhesion performance (Appendix A). The adhesion strength increased with increasing curing time and reached a plateau after 12 h, indicating that the dopamine oxidation would increase the adhesion strength. Moreover, the adhesion performance of PD-L bio-glue prepared using PBS at different pH (4.0, 5.0, 7.4, 9.0, and 10.0) was studied (Appendix A). The adhesion strength first increased and then decreased with the increase in pH, suggesting that the bio-glue was suitable for adhesion at neutral conditions. All these results suggested the fast and strong adhesion of PD-L bio-glue for both organic and inorganic substrates.

An advantage of the PD-L bio-glue based on pectin-dopa and β-lactoglobulin is that the aggregative state can even remain in solution, suggesting that the PD-L bio-glue can work directly in aqueous environments. As shown in Figure 3G, the PD-L bio-glue was added to the surface of the porcine skin underwater, and another porcine skin was covered to the surface to sandwich the glue. After curing for 12 h using a binder clip, the adhesion strength was measured using a standard lap shear test in pure water (ddH_2_O). The measured shear strength of the long-term adhesion reached 0.78 MPa, indicating the excellent adhesion performance of the PD-L bio-glue underwater. The phase separation behavior prevents the diffusion of glue to the aqueous environment during the adhesion operation and curing process, leading to strong adhesion in aqueous environments.

Then, the short- and long-term adhesion strength for adhesion of various tissues (porcine liver, heart, kidney, stomach, and lung) using the PD-L bio-glue was also studied. As shown in Figure 4A–C, the shear strength of short-term adhesion was more than 20 kPa for all tissues (22.1 kPa for liver, 25.5 kPa for the heart, 31.2 kPa for the kidney, 19.4 kPa for the stomach, and 21.8 kPa for the lung), and the shear strength of long-term adhesion was more than 270 kPa (349 kPa for liver, 398 kPa for the heart, 381 kPa for the kidney, 274 kPa for the stomach, and 368 kPa for the lung). The shear strengths of the PD-L bio-glue for different tissues were significantly higher than those of fibrinogen-based bio-adhesives, which are usually less than 30 kPa [8]. The tensile strength of short-term adhesion was higher than 10 kPa for all tissues (16.1 kPa for liver, 15.5 kPa for the heart, 12.2 kPa for the kidney, 20.4 kPa for the stomach, and 17.8 kPa for the lung), and the tensile strength of long-term adhesion was higher than 270 kPa (319 kPa for liver, 308 kPa for the heart, 291 kPa for the kidney, 274 kPa for the stomach, and 248 kPa for the lung) (Figure 4D–F). All these results suggested the considerable instant and long-term adhesion strength of the PD-L glue on various wet tissues.

### 2.4. Regulation of the Adhesion Strength and Biocompatibility of PD-L Bio-Glue

The strong adhesion strength of PD-L bio-glue can be attributed to the adhesion and cohesion of bio-glue. In our bio-glue, cohesion is achieved by the phase separation of pectin and β-lactoglobulin, which can remain aggregative in solution. During adhesion, noncovalent interactions between dopa and substrates, such as ionic interactions, cation–π interactions, and hydrogen bonds, form in seconds. Moreover, dopa can also form covalent bonds with amido/thiol from organic tissues, and amine/thiol covalent bonds ensure long-term strong surface bonding. The combined strong surface bonding and the strength of the bio-glue ensure the overall shear and tensile strength of the bio-glue tapes over megapascal.

Since the adhesion and cohesion of PD-L bio-glue are correlated with the modification of dopa and phase separation, the adhesion strength can be regulated by the synthesis ratios of dopa and pectin or the pectin-dopa:β-lactoglobulin ratios. As shown in Figure 5A and Appendix A, the adhesion strength of PD-L bio-glues for long-term adhesion decreased with the synthesis ratios of dopa and pectin. The shear strength of short-term adhesion for porcine skin decreased from 0.27 MPa at 1:1 to 0.09 MPa at 1:10, while that of long-term adhesion decreased from 1.48 MPa at 1:1 to 0.97 MPa at 1:10 (Figure 5A). Similar trends were observed for the adhesion of glass–glass and skin–glass substrates, indicating that the adhesion strength of PD-L bio-glue increased with the dopa content in pectin-dopa. Moreover, the mass ratio of pectin-dopa and β-lactoglobulin also affected the adhesion strength. The shear strength of the long-term adhesion for porcine skin increased from 0.81 MPa at 1:9 to 1.48 MPa at 5:5 and then decreased to 0.36 MPa at 9:1 (Figure 5B and Appendix A). Similar trends were observed for the short-term adhesion of porcine skin and the adhesion of glass–glass and skin–glass substrates. The highest adhesion strength was achieved at a ratio of 5:5, consistent with the trend of phase separation at various mass ratios of pectin-dopa and β-lactoglobulin. Finally, the cytotoxicity of the bio-glue was evaluated with HeLa and hMSC cells as the model cells. After 24 h of cell culture, almost no dead cells were found in the presence of PD-L bio-glue (Figure 5C and Appendix A). The cell viabilities remained more than 80%, indicating the outstanding biocompatibility of the bio-glue (Figure 5D).

## 3. Materials and Methods

### 3.1. Materials

Pectin was purchased from Shanghai Meryer Chemical Technology Co., Ltd. (Shanghai, China). Dopamine, N-hydroxy succinimide (NHS), and 1-(3-dimethylaminopropyl)-3-ethylcarbodiimide hydrochloride (EDC) were purchased from Sigma-Aldrich (Shanghai, China). The β-Lactoglobulin was purchased from Shanghai Macklin Biochemical Co., Ltd. Unless otherwise stated, all other reagents were purchased from Sinopharm Chemical Reagent Co., Ltd. (Beijing, China). hMSCs and HeLa cells were provided by the Cell Bank of the Chinese Academy of Sciences (Chinese Academy of Sciences, Shanghai, China). 

### 3.2. Synthesis of Pectin-Dopa

The synthesis of pectin-dopa was achieved by the reaction of pectin and dopamine under the catalysis of EDC and NHS. First, pectin and dopamine were dissolved in Milli-Q water to a concentration of 50 mg mL^−1^. Then, EDC and NHS were added to the mixture to concentrations of 50 and 25 mg mL^−1^, respectively. After removing the dissolved oxygen using nitrogen, the reaction was conducted for 12 h in an ice bath with mild stirring. The unreacted reactants were removed by dialysis against excess Milli-Q water using dialysis tubing (Zhuyan Biotech, Nanjing, China) with a molecular weight cut-off of 3.5 kDa. The final product was lyophilized and stored at 4 °C. The linking efficiencies were confirmed by UV spectroscopy (Appendix A). The mass ratios of dopa and pectin were 1:1, 1:2, and 1:10, respectively. 

### 3.3. Preparation of the Bio-Glue

Pectin-dopa and β-lactoglobulin, at a concentration of 250 mg mL^−1^, were prepared using PBS (10 mM, pH = 7.4) and then mixed at a volume ratio of 1:1 to prepare the PD-L glue. The glue was used in all experiments unless otherwise stated. To adjust the adhesion performance, PD-L glue at various mass ratios of pectin-dopa and β-lactoglobulin was prepared. The mass ratios of pectin-dopa and β-lactoglobulin were set as 1:9, 3:7, 5:5, 7:3, and 9:1, with the total mass concentration of pectin-dopa and β-lactoglobulin remaining at 250 mg mL^−1^. For the P-L bio-glue, the concentrations of pectin and pectin were both 125 mg mL^−1^. For the PD bio-glue, the concentration of pectin-dopa was 250 mg mL^−1^.

### 3.4. Dynamic Light Scattering (DLS) Measurement

The mixture of pectin-dopa and β-lactoglobulin was dissolved into PBS solutions (10 mM, pH = 7.4) to a concentration of 0.4 mg mL^−1^. Then, the diameters of the coacervates formed by pectin-dopa and β-lactoglobulin were studied using a dynamic light scattering instrument (Zetasizer Nano-ZS90, Malvern, UK). All measurements were performed for 120 s with multiple scattering suppression at room temperature and a scattering angle of 90°.

### 3.5. Light Microscopic Images

Micrographs were obtained using a fluorescence microscope OLYMPUS-IX73 (OLYMPUS, Tokyo, Japan). Typically, the solution of the mixture of pectin-dopa and β-lactoglobulin was prepared using PBS buffer (10 mM, pH = 7.4) and incubated for 30 min. The total mass concentration of pectin-dopa and β-lactoglobulin remained at 250 mg mL^−1^. Then, the solution was dropped on the glass slices and imaged using a microscope at room temperature.

### 3.6. UV, FT-IR, and ^1^H-NMR Spectroscopy Measurements

The linking efficiency of the pectin-dopa was monitored on a UV/Vis spectrophotometer (Jasco V-550) using a quartz cuvette with a path length of 1 mm. Dopa and pectin solutions at different concentrations were prepared and measured. Pectin-dopa synthesized at different ratios of dopa and pectin was dissolved in ddH_2_O to a concentration of 0.5 mg mL^−1^, and UV spectroscopy was performed.

FTIR spectra were recorded on Bruker VERTEX 80V spectrometers using lyophilized samples. The ^1^H NMR spectra were recorded on a Bruker 500 MHz NMR spectrometer in deuteroxide (D_2_O). The ^1^H NMR spectra of the polymers were merged from more than 128 scans to improve the signal-to-noise ratio.

### 3.7. Adhesion Strength Test

Tissue samples were covered with PBS and stored in plastic bags at 4 °C before measurement. The fat under the porcine skins was removed with a blade, and the dermal surface of the porcine skin was cleaned with alcohol and gauze before the experiments. All the samples were prepared with bio-glue (20 µL, 250 mg mL^−1^) sandwiched between the tissues, and adhesion measurements were performed after pressing for 1 min or 12 h at a pressure of approximately 1.5 kPa. All adhesion measurements were performed using a mechanical testing machine (2 kN load cell, Instron 5944, Norwood, MA, USA).

For the lap shear strength measurements, the width and length of the adhesion areas were approximately 8 mm and 8 mm, respectively. All experiments were conducted in air at 25 °C with a constant stretching rate of 50 mm min^−1^. The shear strength was determined as the maximum stress during the lap shear progress. For the lap shear measurements underwater, the adhesion and curing of the samples were performed in ddH_2_O.

For the tensile strength measurements, the width and length of the adhesion area were approximately 8 mm and 8 mm, respectively. All experiments were conducted in air at 25 °C with a constant stretching rate of 5 mm min^−1^. The tensile strength was determined as the maximum stress during the tensile stretching process.

For the interfacial toughness measurements, the samples were adhered with a width of 10 mm and tested by standard 180-degree peel tests. All experiments were conducted in air at 25 °C with a constant stretching rate of 5 mm min^−1^. The interfacial toughness was determined as two times the maximum force divided by the width of the adhesion area.

### 3.8. Cell Culture and Test

hMSCs were cultured under the conditions: at 37 °C and 5% CO_2_ atmosphere with α-minimum essential medium, 10% fetal bovine serum, 1% streptomycin/penicillin, and 1% L-glutamine (all from Thermo Scientific, Waltham, MA, USA). HeLa cells were cultured under the conditions: at 37 °C and 5% CO_2_ atmosphere with Dulbecco’s modified Eagle’s medium, 10% fetal bo-vine serum, 1% streptomycin/penicillin, and 1% L-glutamine (all from Thermo Scientific). The medium was changed every 3~4 days.

For the cell culture in bio-glue, hMSCs and HeLa cells were trypsinized using 0.05% trypsin (Invitrogen, Shanghai, China) and washed using serum-free DMEM. Then, the cells were seeded at a density of 2 × 10^4^ mL^−1^ in the presence of bio-glue (50, 100, 150, 200, and 250 mg mL^−1^). The incubation of cells was performed at 37 °C and 5% CO_2_. After 24 h, cytotoxicity of the bio-glue was evaluated using a live/dead assay (Calcein-AM/PI Double Staining Kit).

## 4. Conclusions

In summary, we reported glue-type bio-adhesives suitable for the rapid and strong adhesion of tissues based on the phase separation of pectin and β-lactoglobulin. By modifying pectin with the adhesive group (dopa), the bio-glue achieved enhanced surface adhesion abilities while the phase separation remained. Due to the cohesion of phase separation and adhesion of dopa, the bio-glue exhibited excellent instant and long-term adhesion performance for both organic and inorganic substrates. Moreover, bio-glue can even work normally in aqueous environments due to phase separation. Finally, we demonstrated the feasibility of this glue-type bio-adhesive in the adhesion of various visceral tissues as well as its excellent biocompatibility. Given the convenience of application, biocompatibility, and robust bio-adhesion, we anticipate the bio-glue may find broad biomedical and clinical applications. The principle used in this work also represented a general approach to build glue-type tissue adhesives based on proteins and polysaccharides.

## Figures and Tables

**Figure 1 ijms-23-09987-f001:**
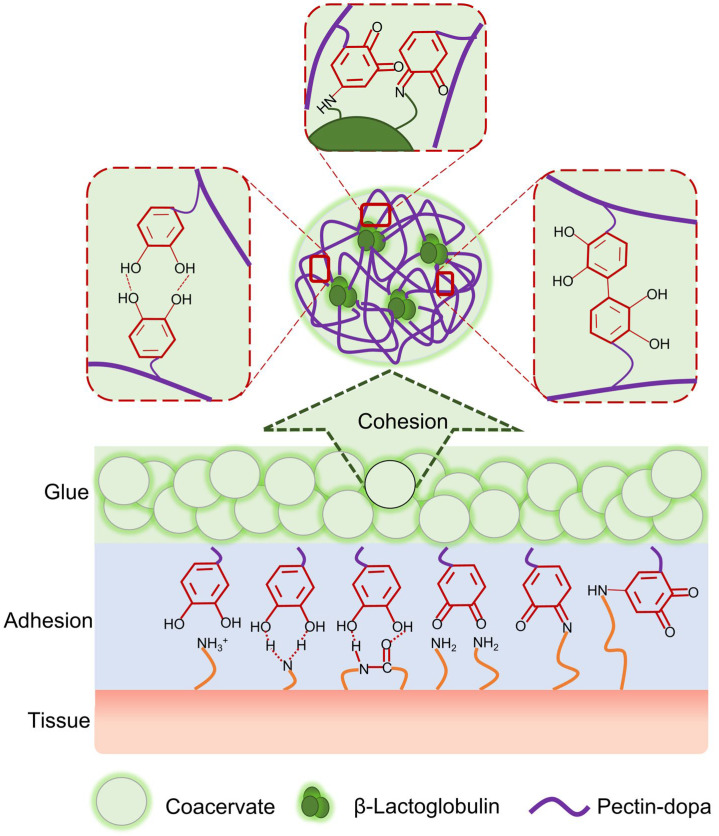
Schematic of the bio-glue based on the liquid–liquid phase separation of dopa-modified pectin and β-lactoglobulin. The bio-glue is formed by dopa-modified pectin and β-lactoglobulin. Cohesion in bio-glue at the initial state is achieved by the phase separation of pectin and β-lactoglobulin, which can even keep the aggregate state in solution. The adhesion of the bio-glue is obtained by the introduction of dopa into pectin. During adhesion, noncovalent interactions between dopa and the substrate, such as ionic interactions, cation–π interactions, and hydrogen bonds, form in seconds. Furthermore, dopa can also form covalent bonds with amido/thiol from organic tissues, and the adhesion strength is gradually enhanced in hours.

**Figure 2 ijms-23-09987-f002:**
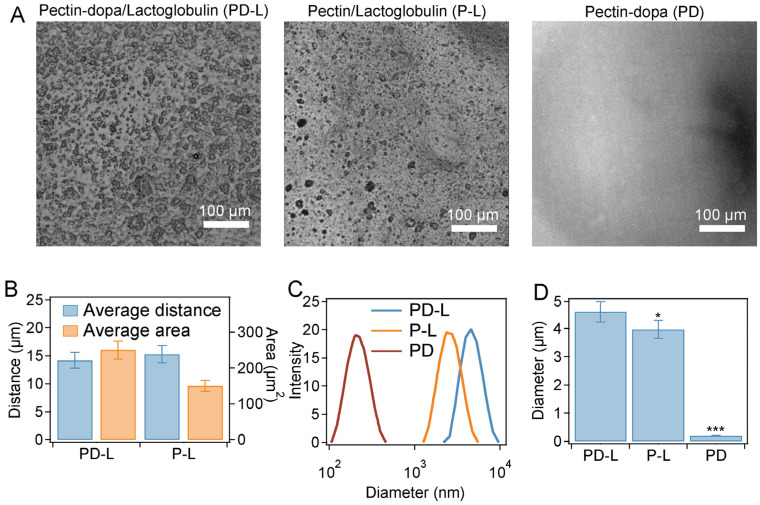
Phase separation of pectin-dopa and β-lactoglobulin. (**A**) Micrographs of the mixture of pectin-dopa and β-lactoglobulin (PD-L), pectin-dopa and β-lactoglobulin (P-L), and pectin-dopa (PD) in PBS (10 mM, pH = 7.4). (**B**) Average distance between the adjacent coacervates and average area of the coacervates. (**C**) Coacervate size distributions of the PD-L, P-L, and PD bio-glue measured using DLS. (**D**) Average diameters of the coacervates in the PD-L, P-L, and PD bio-glue measured using DLS. Error bars represent SD, and asterisks denote statistical significance compared with PD-L bio-glue (*p* < 0.05: *; *p* < 0.001: ***).

**Figure 3 ijms-23-09987-f003:**
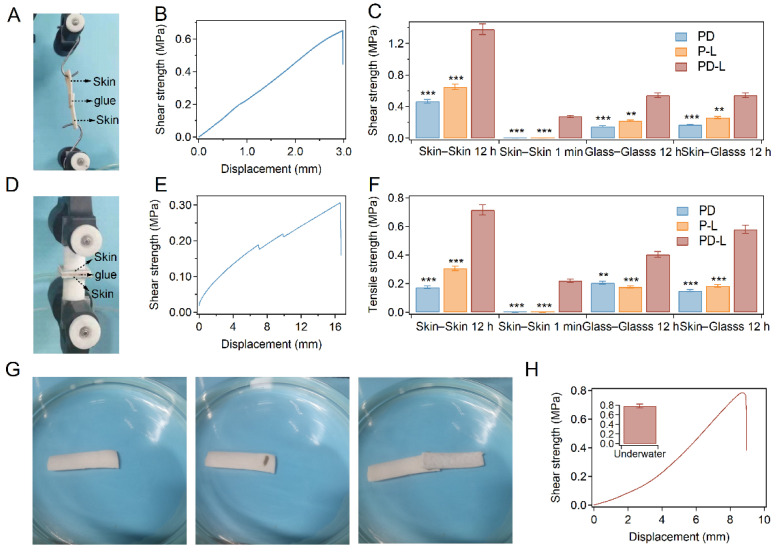
Adhesion performance of the bio-glue. (**A**,**B**) Typical image (**A**) and force–displacement curve (**B**) recorded in the lap shear test. The lap shear strength of the bio-glue was determined as the fracture strength in the lap shear test. (**C**) Shear strength of short-term (1 min) and long-term (12 h) adhesion using PD, P-L, and PD-L bio-glues. Error bars represent SD. (**D**,**E**) Typical image (**D**) and force–displacement curve (**E**) recorded in the tensile strength test. The tensile strength of the bio-glue was determined as the fracture strength in the tensile strength test. (**F**) Tensile strength of short-term (1 min) and long-term (12 h) adhesion using PD, P-L, and PD-L bio-glues. Error bars represent SD. (**G**) Typical underwater adhesion images. Scale bar = 10 mm. (**H**) Typical force–displacement curve recorded in the lap shear test for long-term (12 h) skin–skin adhesion underwater. Asterisks denote statistical significance compared with PD-L bio-glue (*p* < 0.01: **; *p* < 0.001: ***).

**Figure 4 ijms-23-09987-f004:**
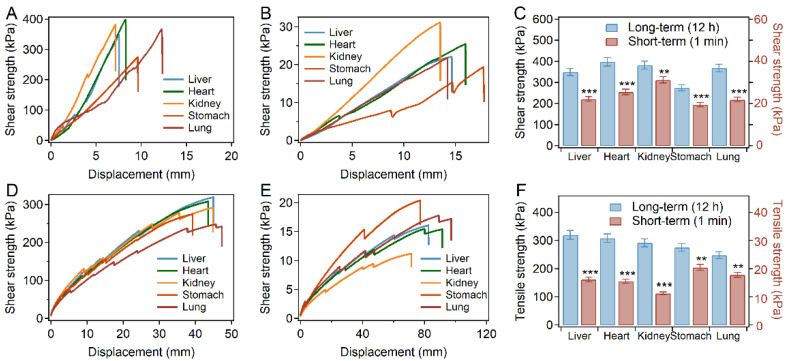
Adhesion performance of the bio-glue on different organic tissues. (**A**,**B**) Typical force–displacement curves recorded in the long-term (**A**) and short-term (**B**) adhesion measurements of shear strength on different tissues. (**C**) Shear strength of short-term (1 min) and long-term (12 h) adhesion of the PD-L bio-glue on different tissues. Error bars represent SD. (**D**,**E**) Typical force–displacement curves recorded in the long-term (**D**) and short-term (**E**) adhesion measurements of tensile strength on different tissues. (**F**) Tensile strength of short-term (1 min) and long-term (12 h) adhesion of the PD-L bio-glue on different tissues. Error bars represent SD. Asterisks denote statistical significance compared with long-term adhesion (*p* < 0.01: **; *p* < 0.001: ***).

**Figure 5 ijms-23-09987-f005:**
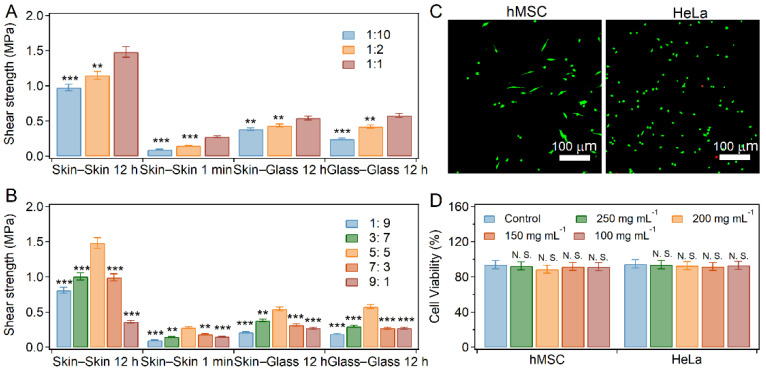
Regulation of the adhesion strength and biocompatibility of PD-L bio-glue. (**A**) Adhesion performance of bio-glue at different synthesis ratios of dopa and pectin. Error bars represent SD. Asterisks denote statistical significance compared with bio-glue at a ratio of 1:1. (**B**) Adhesion performance of bio-glue at different mass ratios of pectin-dopa and β-lactoglobulin. Error bars represent SD. Asterisks denote statistical significance compared with bio-glue at a ratio of 5:5. (**C**) Fluorescence microscopic images of hMSC (left) and HeLa (right) cells cultured in the presence of bio-glue (250 mg mL^−1^). The living (green) and dead (red) cells were stained with a live/dead assay (Calcein-AM/PI Double Staining Kit) after 24 h of culture. (**D**) Cell viabilities of the HeLa and hMSC cells cultured in the presence of bio-glue (50, 100, 150, 200, and 250 mg mL^−1^). Error bars represent SD. Asterisks denote statistical significance compared with control (*p* > 0.05: N.S.; *p* < 0.01: **; *p* < 0.001: ***).

## Data Availability

All data are available in the main text or the Appendix A.

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
