# Peer review of "Engineering Bio-Adhesives Based on Protein–Polysaccharide Phase Separation"

_ijms, 2022, doi:10.3390/ijms23179987_

Round 1

Reviewer 1 Report

In this paper, the authors presented a bio-adhesives using the synergic effect of phase separation and dihydroxyl phenylalanine. The results show that the adhesion performance of this type of glue is significantly superior to the control samples. The biocompatibility of the glue was also tested and the result is promising. I recommend the publication of the study in its current form except a typo In Figure 2D. The unit of the diameter is wrong.

Author Response

Reviewer 1:

In this paper, the authors presented a bio-adhesives using the synergic effect of phase separation and dihydroxyl phenylalanine. The results show that the adhesion performance of this type of glue is significantly superior to the control samples. The biocompatibility of the glue was also tested and the result is promising. I recommend the publication of the study in its current form except a typo In Figure 2D. The unit of the diameter is wrong.

Response:We thank the reviewer for the positive comments. Now we have corrected the unit of Figure 2D according to the reviewer’s comment. The revised Figure 2 is attached as below. (See Figure 2 on Page 5 in the revised manuscript)

Revision:

Figure 2. Phase separation of pectin-dopa and β-lactoglobulin. (A) Micrographs of the mixture of pectin-dopa and β-lactoglobulin (PD-L), pectin-dopa and β-lactoglobulin (P-L), and pectin-dopa (PD) in PBS (10 mM, pH=7.4). (B) Average distance between the adjacent coacervates and average area of the coacervates. (C) Coacervate size distributions of the PD-L, P-L and PD bio-glue measured using DLS. (D) Average diameters of the coacervates in the PD-L, P-L and PD bio-glue measured using DLS. Error bars represent SD, and asterisks denote statistical significance compared with PD-L bio-glue (P > 0.05: N.S.; P < 0.05: *; P < 0.01: **; P < 0.001: ***).

Reviewer 2 Report

The authors have reported the new type of bio-glue composed of DOPA-modified pectin and β-lactoglobulin based on the phase separation of pectin and β-lactoglobulin for using adhesion of tissues. These results will be helpful and informative for researchers in the field of materials science and materials chemistry for biomaterials. Whereas the reviewer thinks that the authors’ study in this manuscript is quite interesting, suggestive, and well-organized, some descriptions are not enough. The authors’ manuscript is not suitable for publication in “International Journal of Molecular Science” in the present form.

From these considerations, the reviewer recommends to accepting for publication in " International Journal of Molecular Science," if the following issues are resolved.

(1)   How did the authors verify that the interactions described in Figure 1 exist in this study? The reviewer thinks that the experimental results and discussion of these molecular interactions depicted in Figure 1 are insufficient. Is there a reference or prior study about Figure 1 that the author can point to? It seems to the reviewer that infrared absorption spectroscopy is a useful tool for quantitatively estimating the presence or absence and the degree of contribution of the intermolecular interactions schematically shown in Figure 1.

(2)   Is there any correlation between the results of adhesion performance of bio-glue and various intermolecular interactions shown in Figure 1? 

Author Response

(1)   How did the authors verify that the interactions described in Figure 1 exist in this study? The reviewer thinks that the experimental results and discussion of these molecular interactions depicted in Figure 1 are insufficient. Is there a reference or prior study about Figure 1 that the author can point to? It seems to the reviewer that infrared absorption spectroscopy is a useful tool for quantitatively estimating the presence or absence and the degree of contribution of the intermolecular interactions schematically shown in Figure 1.

Response: We thank the reviewer for the comments. The interactions illustrated in Figure 1 have been widely studied and reported by researcher all over the world. We and others also have investigated and reported the mechanism of dopa-based adhesions and polymerization. (Nat Commun 11, 3895 (2020); Nanoscale Adv. 1, 4246–4257 (2019); J. Polym. Sci. B Polym. Phys. 49, 757–771 (2011).) We have citied more references following the reviewer’s comments. (See reference No. 49-51 on page 3 in the revised manuscript)

Moreover, we also investigated the infrared absorption spectroscopy of the bio-glue to study the intermolecular interactions. As shown in Figure S4, the peak at 3200-3500 cm-1 enhanced in the PD-L bio-glue comparing to that of P-L bio-glue, indicating the hydrogen bond and dopa oxidations. The new comments and Figure S4 are included in the revised manuscript now. (See Line 147-150 on page 4 in the revised manuscript and Figure S4 in the revised supplementary materials)

Revision:

The fourier transform infrared (FT-IR) spectroscopy of PD-L, P-L and PD were shown in Figure S2. The peaks at 3200-3500 cm-1 enhanced in the PD-L bio-glue comparing to that of P-L bio-glue, indicating the hydrogen bond and dopa oxidations.

Figure S4. Fourier transform infrared (FT-IR) spectroscopy of the PD, P-L and PD-L bio-glues.

(2)   Is there any correlation between the results of adhesion performance of bio-glue and various intermolecular interactions shown in Figure 1? 

Response: We thank the reviewer for the comments. According to the intermolecular interactions illustrated in Figure 1, the cohesion and adhesion of PD-L bio-glue would enhance with the oxidation of dopa. As a result, the adhesion strength would increase with the oxidation of dopa. So, we studied the adhesion strength of the PD-L bio-glue on porcine skin at different oxidation times. As expected, the adhesion strength increased with the oxidation time (Figure S10). The adhesion strength increased with increasing curing time and reached a plateau after 12 h, consistent with the prediction. We have included new data and comments in the revised manuscript. (See Line 191-195 on page 5 in the revised manuscript and Figure S10 in the revised supplementary materials)

Revision:

The adhesion strength after curing for different times (1 min, 1, 3, 6, 12, 18 and 24 h) was also studied to investigate the effects of dopamine oxidation kinetic on the adhesion performance (Figure S10). The adhesion strength increased with increasing curing time and reached a plateau after 12 h, indicating that the dopamine oxidation would increase the adhesion strength.

Figure S10. Adhesion strength of PD-L bio-glue at different oxidation times. (A) Typical force–displacement curves recorded in the lap shear experiment for the porcine skin adhesion using PD-L bio-glue after curing for different times. (B) Adhesion strength for the porcine skin using PD-L bio-glue after curing for different times. Error bars represent SD.

Reviewer 3 Report

The manuscript describes the production of an injectable glue-like bio-adhesive based on pectin-dopa and β-lactoglobulin. The results are interesting especially because they demonstrate a synergic effect of coacervation and adhesion properties of dopamine.

Below some issue/comments that I suggest to address before publication:

1)    Since authors stated that the obtained bio-adhesive is injectable. It would be interesting to study the rheological properties of the obtained glues. Authors should specify how this system could be prepared and used in the clinic.

2)    Paragraph 3.6 could be merged with paragraph 3.3. Also, authors should specify what ddH2o is

3)    Statistical analysis must be reported for all the experiments

4)    Explain the reason why, for the cytocompatibility experiments, the bio glue concentration was changed compared to the adhesion experiments. in my opnion it should be interesting to test the cytocompatibility at different concentration.

5)    Since the adhesive properties at 1 min are comparable for all the investigated sample (glue without dopa and pectine-dopa alone) it might be interesting to investigate the dopamine oxidation kinetic at the experimental conditions used for the adhesion tests? Is it possible that changing the medium parameters (pH as exmple) leads to completely different adhesion behavior?

6)    It should be interesting to report 1H-NMR of the obtained pectine-dopa products along with UV spectra.

Author Response

(1) Since authors stated that the obtained bio-adhesive is injectable. It would be interesting to study the rheological properties of the obtained glues. Authors should specify how this system could be prepared and used in the clinic.

Response: We thank the reviewer for the comments. Now we have studied the rheological properties of the bio-glue following the reviewer’s comments. As shown in Figure S3A-C, the storage modulus (G’) and loss modulus (G”) of the P-L bio-glue were 2410 and 1726 Pa, and increased with the increasing frequency, indicating a typical response of polymers. In contrast, G’ and G’’ of P-L bio-glue were 694 and 237 Pa, and slightly increased in the frequency range of 1-10 Hz, suggesting enhanced dynamic stabilities. The high G’ and stable response to frequency of PD-L bio-glue indicated the enhanced cohesion inside the PD-L bio-glue, which is consistent with the observation of phase separation. G’ and G” of pectin-dopa (G’ ~75 Pa; G’’ ~37 Pa) was much smaller than those of PD-L and P-L, indicating the weaker cohesion.

Moreover, the PD-L bio-glue show a shear thinning property with the increase of frequency in the rheological experiments (Figure S3D). Because of the shear-thinning properties, the bio-glue can be injected to the tissue defects and curing the wound locally in the clinic. The new data and comments are included in the revised manuscript. (See Line 136-147 on page 4 in the revised manuscript and Figure S3 in the revised supplementary materials)

Revision:

Furthermore, the rheological properties of the bio-glues were investigated. As shown in Figure S3A-C, the storage modulus (G’) and loss modulus (G”) of the P-L bio-glue were 2410 and 1726 Pa, and increased with the increasing frequency, indicating a typical response of polymers. In contrast, G’ and G’’ of P-L bio-glue were 694 and 237 Pa, and slightly increased in the frequency range of 1-10 Hz, suggesting enhanced dynamic stabilities. The high G’ and stable response to frequency of PD-L bio-glue indicated the enhanced cohesion inside the PD-L bio-glue, which is consistent with the observation of phase separation. G’ and G” of pectin-dopa (G’ ~75 Pa; G’’ ~37 Pa) was much smaller than those of PD-L and P-L, indicating the weaker cohesion. Moreover, the PD-L bio-glue show a shear thinning property with the increase of frequency (Figure S3D), suggesting that the bio-glue can be injected to the tissue defects directly in the clinic.

Figure S3. Rheological properties of the bio-glues. (A-C) G’ and G” of pectin-dopa (A), P-L (B) and PD-L (C) bio-glue with ion measured in a frequency sweep experiment (from 0.01 to 90 Hz, 1% strain) at room temperature. (D) Viscosity measurement of PD-L bio-glue at 20 °C.

2)    Paragraph 3.6 could be merged with paragraph 3.3. Also, authors should specify what ddH2o is

Response:We thank the reviewer for the comments. ddH2O is the pure water. Now we have specified the ddH2O in the revised manuscript. (See Line 207 on Page 6 in the revised manuscript)

3)    Statistical analysis must be reported for all the experiments

Response:We thank the reviewer for the comments. Now statistical analysis has been reported for all the experiments. (See Figures 2-5 in the revised manuscript and Figures S5-6 in the revised supplementary materials)

4)    Explain the reason why, for the cytocompatibility experiments, the bio glue concentration was changed compared to the adhesion experiments. in my opnion it should be interesting to test the cytocompatibility at different concentration.

Response: We thank the reviewer for the comments. Now we have performed the cytocompatibility experiments at different concentrations of PD-L bio-glue (50, 100, 150, 200 and 250 mg mL-1). The cell viability of cells remained higher than 90% at various concentrations (Figure 5D and S14), indicating the excellent cytocompatibility of the PD-L bio-glue. The new data and comments are included in the revised manuscript. (See Figure 5C-D on Page 8 in the revised manuscript and Figure S13 in the revised supplementary materials)

Revision:

Figure 5. Regulation of the adhesion strength and biocompatibility of PD-L bio-glue. (A) Adhesion performance of bio-glue at different synthesis ratios of dopa and pectin. Error bars represent SD. Asterisks denote statistical significance compared with bio-glue at a ratio of 1: 1. (B) Adhesion performance of bio-glue at different mass ratios of pectin-dopa and β-lactoglobulin. Error bars represent SD. Asterisks denote statistical significance compared with bio-glue at a ratio of 5: 5. (C) Fluorescence microscopic images of hMSC (left) and HeLa (right) cells cultured in the presence of bio-glue (250 mg mL-1). The living (green) and dead (red) cells were stained with a live/dead assay (Calcein-AM/PI Double Staining Kit) after 24 h of culture. (D) Cell viabilities of the HeLa and hMSC cells cultured in the presence of bio-glue (50, 100, 150, 200 and 250 mg mL-1). Error bars represent SD. Asterisks denote statistical significance compared with control (P>0.05: N.S.; P < 0.05: *; P < 0.01: **; P < 0.001: ***).

Figure S14. Fluorescence microscopic images of HeLa (left) and hMSC (right) cells cultured in the presence of bio-glue (50, 100, 150, 200 and 250 mg mL-1 and control group. The living (green) and dead (red) cells were stained with a live/dead assay (Calcein-AM/PI Double Staining Kit) after 24 h of culture.

5)    Since the adhesive properties at 1 min are comparable for all the investigated sample (glue without dopa and pectine-dopa alone) it might be interesting to investigate the dopamine oxidation kinetic at the experimental conditions used for the adhesion tests? Is it possible that changing the medium parameters (pH as exmple) leads to completely different adhesion behavior?

Response:We thank the reviewer for the comments. Dopaminequinone, one of the oxidation products of dopa, can form covalent bonding with thiol/amino on the tissue face. As a result, the adhesion strength would enhance with the oxidation time (curing time). Following the reviewer’s comments, we studied the adhesion strength after curing for different times (1 min, 1, 3, 6, 12, 18 and 24 hours) to investigate the effects of dopamine oxidation kinetic on the adhesion performance (Figure S10). As expected, the adhesion strength for the porcine skins increased with the curing time, indicating that the dopamine oxidation would increase the adhesion strength. Moreover, the adhesion performance of PD-L bio-glue prepared using PBS at different pH (3.0, 5.0, 7.4, 9.0 and 10.0) was studied. As shown in Figure S11, the adhesion strength first increased and then decreased with the increase of pH, suggesting that the bio-glue was suitable for adhesion at neutral conditions.

Revision:

The adhesion strength after curing for different times (1 min, 1, 3, 6, 12, 18 and 24 h) was also studied to investigate the effects of dopamine oxidation kinetic on the adhesion performance (Figure S10). The adhesion strength increased with increasing curing time and reached a plateau after 12 h, indicating that the dopamine oxidation would increase the adhesion strength. Moreover, the adhesion performance of PD-L bio-glue prepared using PBS at different pH (4.0, 5.0, 7.4, 9.0 and 10.0) was studied (Figure S11). The adhesion strength first increased and then decreased with the increase of pH, suggesting that the bio-glue was suitable for adhesion at neutral conditions.

Figure S10. Adhesion strength of PD-L bio-glue at different oxidation times. (A) Typical force–displacement curves recorded in the lap shear experiment for the porcine skin adhesion using PD-L bio-glue after curing for different times. (B) Adhesion strength for the porcine skin using PD-L bio-glue after curing for different times. Error bars represent SD.

Figure S11. Adhesion strength of PD-L bio-glue at different pH. (A) Typical force–displacement curves recorded in the lap shear experiments for the long-term porcine skin adhesion using PD-L bio-glue prepared using PBS at different pH (4.0, 5.0., 7.4, 9.0 and 10.0). (B) Adhesion strength for the long-term porcine skin adhesion using PD-L bio-glue prepared using PBS at different pH. Error bars represent SD.

6)    It should be interesting to report 1H-NMR of the obtained pectine-dopa products along with UV spectra.

Response:We thank the reviewer for the comments. Following the reviewer’s comment, we have reported 1H-NMR of pectin-dopa (Figure S2). As indicated by the chemical shift at 6.5-6.9 which belongs to H from benzene, dopa was successfully connected with pectin. 1H-NMR and the new comment are included in the revised manuscript. (See Figure S2 on Page 1 in the revised supplementary materials and Line 122-124 on Page 4 in the revised manuscript)

Revision:

The chemical shift form H nuclear magnetic resonance (1H-NMR) spectrum at 6.5-6.9 which belongs to H of benzene further indicated that dopa was successfully connected with pectin (Figure S2).

Figure S2. H Nuclear Magnetic Resonance (1H-NMR) spectroscopy of dopa, pectin and pectin-dopa
